# Dosage Optimisation of Trimethoprim and Sulfamethoxazole for the Treatment of an Avian Pathogenic Strain of *Escherichia coli* in Broiler Chickens

**DOI:** 10.3390/antibiotics13010011

**Published:** 2023-12-20

**Authors:** Kamil Stastny, Nikola Hodkovicova, Martin Jerabek, Michal Petren, Michaela Viskova, Aneta Papouskova, Iva Bartejsova, Kristina Putecova-Tosnerova, Michaela Charvatova, Monika Zouharova, Katarina Matiaskova, Katerina Nedbalcova

**Affiliations:** 1Veterinary Research Institute, Hudcova 296/70, 621 00 Brno, Czech Republic; kamil.stastny@vri.cz (K.S.); nikola.hodkovicova@vri.cz (N.H.); iva.bartejsova@vri.cz (I.B.); kristina.tosnerova@vri.cz (K.P.-T.); michaela.charvatova@vri.cz (M.C.); monika.zouharova@vri.cz (M.Z.); katarina.matiaskova@vri.cz (K.M.); 2Tekro, Ltd., Visnova 484/2, 140 00 Prague, Czech Republic; m.jerabek@tekro.cz (M.J.); m.petren@tekto.cz (M.P.); m.viskova@tekro.cz (M.V.); 3Institute of Infection Diseases and Microbiology, Faculty of Veterinary Medicine, University of Veterinary Sciences, Palackeho 1-3, 612 42 Brno, Czech Republic; papouskovaa@vfu.cz

**Keywords:** poultry, *Escherichia coli*, APEC, antimicrobials, LC-MS/MS, pharmacokinetics

## Abstract

Based on pharmacokinetic studies carried out according to the methodologies defined by the European Medicines Agency (EMA) using mass spectrometry analysis, a new formulation of a veterinary drug for the treatment of broiler chickens is proposed. Currently, the traditional trimethoprim–sulfamethoxazole drug used for broilers is applied in a 1:5 ratio, and the recommended dose is 45 mg kg^−1^ of live weight administered at 24 h intervals for 3 to 5 days. In this study, we propose a novel combination containing similar active substances in a newly established ratio of 1:4, with a recommended dosage of 20 mg kg^−1^ of live weight administered at 24 h intervals for 3 to 5 days. With this method, the currently recommended dose of the traditional trimethoprim–sulfamethoxazole drug used for broilers can be reduced by more than half. The efficacy of the newly designed formulation and dosage of the drug was verified in a bioassay for the treatment of broilers experimentally infected with an avian pathogenic strain of *Escherichia coli*. In the experiment, we compared the newly designed dosage with the traditional dosage in terms of efficacy and dosage. There were no statistically significant differences between the two drugs in efficacy regarding the survival of chickens after experimental infection or changes in their health status. The experimental results suggest that a significant reduction in the recommended daily dose of drugs containing trimethoprim and sulfamethoxazole for the treatment of bacterial infections in broilers is possible and can support the prudent use of antimicrobials, including the limitation of their overuse.

## 1. Introduction

Avian pathogenic *Escherichia coli* (APEC) is one of the pathotypes of *E. coli* that causes infectious diseases in domestic and wild birds, specifically avian colibacillosis. In poultry farms, these colibacilloses represent some of the most serious diseases that affect poultry of all ages and performance categories [1]. They manifest as various local or systemic infections with a wide range of clinical signs and post mortem pathological and anatomical findings, from acute septicaemia and sudden death as a result of systemic inflammation in different parts of the body, such as coliform peritonitis syndrome (EPS) or salpingo-peritonitis syndrome (SPS), to chronic, mostly respiratory infections [2]. In broiler breeding, navel and yolk sac infections and early mortality are the most common manifestations of infection, followed by air sac inflammation and polyserositis starting from approximately two weeks of age. The severity of the course depends on the virulence of the causative agent and several predisposing factors, such as zoohygiene, stress, and immunosuppression [3]. In the event of a disease outbreak on the farm, the initiation of antibiotic treatment is necessary, as colibacillosis is otherwise challenging to manage [4].

However, the inappropriate or careless use of antimicrobials to treat infections in human and veterinary medicine leads to the emergence of antimicrobial resistance, which has become a problem worldwide in recent years. For this reason, one of the key challenges for contemporary medicine is to maintain the efficacy of the currently used antimicrobials, which is seriously threatened by the increasing prevalence of the resistance of microorganisms to these treatments. A well-defined dosage of antimicrobials based on knowledge of the pharmacokinetic (PK) and pharmacodynamic (PD) properties of the agents, which determine the necessary levels of exposure of antimicrobials for the maximum effect on the bacteria, contributes significantly to limiting the emergence and development of resistance [5].

Currently, fluoroquinolones, especially enrofloxacin, are successfully used to treat poultry colibacillosis due to their broad-spectrum effects and short withdrawal periods [6,7]. However, fluoroquinolones are now category “B” antibiotics according to the European Medicines Agency (EMA) classification, with strict restrictions on their use for animal treatment [8]. As there are currently few products registered for the antibiotic treatment of poultry worldwide, it is necessary to search for alternatives that could replace fluoroquinolones in the future.

Sulfonamides are synthetic compounds derived from sulfanilamide that share a common mode of action but differ greatly in their chemical properties, usual routes of administration, and pharmacokinetics. Their antimicrobial activity results from their ability to inhibit parts of the microbial folic acid pathway, which interferes with DNA synthesis. Sulfonamides are effective against Gram-positive and Gram-negative bacteria, protozoa, and coccidia. Sulfonamides are well absorbed in the gastrointestinal tract, depending on solubility. Absorption occurs more rapidly in birds than in mammals. Sulphonamides are widely distributed in tissues, the main route of excretion in both mammals and birds being the kidneys, although some are excreted in faeces. In poultry, the most common route of administration is per os via water [9].

In many cases, sulfonamides are potentiated by diaminopyrimidine compounds such as trimethoprim, which increase their efficacy. The combination of these two chemotherapeutics has a synergistic antibacterial effect in vitro due to the inhibition of distinct steps in the bacterial folic acid biosynthetic pathway. This synergism not only reduces the minimum inhibitory concentrations of individual substances but also broadens the bacterial spectrum and reduces resistance.

According to the EMA, trimethoprim and sulfamethoxazole are classified in category D “Prudence”; their use should therefore be responsible and considered to keep the risk as low as possible [8]. The purpose of this study is to find an alternative dosage of the preparation containing these substances for broilers in such a way that its effectiveness against bacterial infections is maintained but, at the same time, antibiotic resistance caused by the excessive use of antibiotics is not exacerbated. Moreover, the risks associated with the application of veterinary preparations and the potential risks for consumers of food products of animal origin are specified in Directive 2001/82/EC [9]. To maintain the safety of food products for consumers, it is necessary to maintain the withholding period of veterinary substances, which is possible based on the defined PK properties of the substances.

The available data on the kinetics of sulfamethoxazole and trimethoprim in birds are very limited and suggest that after oral administration, bioavailability may range between 35 and 80%, and these active substances are widely distributed throughout the body [10,11]. Different types of sulfonamide–trimethoprim combinations are used to treat bacterial diseases in poultry. Most commercial preparations contain these active ingredients in a 5:1 ratio. The reasons for choosing such a ratio are not clearly stated in the literature but may be extrapolated from the ratio of these substances used in human medicine. However, this ratio may not be optimal for broilers given the significant differences in the distribution and elimination kinetics of these two substances [12].

The main aim of this study was to investigate and propose an alternative dosage of the drug based on sulfamethoxazole and trimethoprim for the treatment of co-infections in broiler chickens. The aims of our experiment, which is based on preliminary data obtained as part of a long-term project, are in line with the hypothesis formed based on the gained preliminary data, i.e., that the currently recommended dose of the traditional trimethoprim-sulfamethoxazole drug used for broilers can be reduced by more than half while maintaining the same effectiveness. The aims established in this study were as follows:To apply data obtained from our previous study [13] involving the determination of PK parameters and antimicrobial efficacy assessment for the single and repeated oral administration of sulfamethoxazole potentiated with trimethoprim using the recommended dosage for poultry and to establish a new experimental dosage of the traditionally used preparation;To compare the effects on the survival rate of experimentally infected individuals after treatment with an experimentally designed dose of the sulfamethoxazole and trimethoprim combination;To compare the efficacy of the experimentally designed dosage of sulfamethoxazole/trimethoprim with that which is traditionally used.

In summary, in optimising the dosages of antibiotics for broilers, this study can contribute to the more judicious use of antimicrobials in veterinary medicine. This approach not only addresses concerns related to bacterial resistance but also yields positive outcomes in terms of animal welfare, food chain safety, environmental impact, and economic efficiency in poultry farming.

## 2. Results

We hypothesised that the new combination of trimethoprim and sulfamethoxazole would not only demonstrate efficacy in reducing reliance on enrofloxacin and other fluorinated quinolones but also contribute to economic sustainability in poultry farming in terms of the proper handling of bacterial diseases. Original data were obtained using modern mass spectrometry methods to define the PK and PD parameters of trimethoprim and sulfamethoxazole. The efficacy of these antimicrobials was determined in terms of the time dependence of their concentration in the blood of broiler chickens after the single and repeated administration of the drug, and an elimination curve for these agents that is unique to broiler chickens was established [13]. From these data, a new optimal formulation containing a combination of trimethoprim and sulfamethoxazole is proposed, including a description of the treatment, route of administration, duration of treatment, and optimal dosage. The efficacy of this preparation was verified through the treatment of experimentally infected broilers.

### 2.1. Validation and Pharmacokinetics

The analytical method used for the targeted quantification of both drugs in chicken sera was validated. Regarding the performance characteristics, the precision was RSD < 6.2% (RSD—relative standard deviation, n = 18) for both analytes, and the inter-precision was RSD < 7.8% when repeated measurements were performed on three different days. Furthermore, the method’s sensitivity was calculated from a linear regression of the matrix calibration curve and was LOD = 0.058 µg mL^−1^, LOQ = 0.178 µg mL^−1^ for trimethoprim and LOD = 0.022 µg mL^−1^, and LOQ = 0.067 µg mL^−1^ for sulfamethoxazole. The correlation coefficient R was greater than 0.99 in both cases.

The pharmacokinetic (PK) parameters, including the maximum serum concentration (C_max_), the time when the maximum concentration was reached (T_max_), and the area under the pharmacokinetic curve (AUC), were determined within the range of the EU standard Guideline EMEA/CVMP/EWP/133/1999. The summary PK parameters were as follows: after a single p.o. application at a dose of 33 mg kg^−1^ of body weight (b.w.) in a 5:1 ratio (27.5 mg sulfamethoxazole + 5.5 mg trimethoprim), the maximum concentrations of C_max_ = 47.1 ± 15.3 µg mL^−1^ at T_max_ = 1 h for sulfamethoxazole and C_max_ = 2.1 ± 1.0 µg mL^−1^ at T_max_ = 1.5 h for trimethoprim, respectively, were established. The AUC was 94.6 ± 28.4 µg h mL^−1^ and 2.9 ± 1.5 µg h mL^−1^ for sulfamethoxazole and trimethoprim, respectively. After reapplication over 3 days, all the PK parameters were measured in the appropriate range, i.e., C_max_ = 35.6 µg mL^−1^ at T_max_ = 1 h and AUC = 91.1 ± 25.2 µg h mL^−1^ for sulfamethoxazole, with C_max_ = 1.6 µg mL^−1^ at T_max_ = 2.2 h and AUC = 2.71 ± 1.2 µg h mL^−1^ for trimethoprim. The PK curves are presented in Figure 1. Other PK parameters such as the elimination rate constant (Kel) or elimination half-life (T1/2) have been calculated and are clearly presented in our previous work; see Putecova et al. [13].

### 2.2. Antimicrobial Efficacy of Trimethoprim with Sulfamethoxazole

The efficacy of the combination of trimethoprim and sulfamethoxazole was investigated by determining the minimal inhibitory concentrations (MICs) of APEC isolates from farms across the Czech Republic, as shown in Table 1. The division of the isolates into susceptibility categories (susceptible and resistant) was performed according to the interpretation criteria (= breakpoints) given for *E. coli* (*Enterobacterales*) in the European Committee on Antimicrobial Susceptibility Testing (EUCAST) [14]. Susceptible isolates have MICs ranging from ≤0.5/9.5 mg L^−1^ to 1/19 mg L^−1^ and resistant isolates have MICs higher than 2/38 mg L^−1^. The results show that most isolates were sensitive to the combination of trimethoprim and sulfamethoxazole (90 isolates; 79.6%). The MIC_50_ and MIC_90_ were ≤0.5/9.5 mg L^−1^ and >64/1216 mg L^−1^, respectively.

### 2.3. Animal Experiment

The weights and average daily gain (AGD) of individual chickens and mortality among different groups are shown in Figure 2 and Figure 3 (for data see Appendix A). The weight gain of chicks in the experimentally infected groups (i.e., groups 1 to 3) was comparable from the time of experimental infection to the end of the experiment. In group 1, which received the experimental dosage, one chicken exhibiting clinical signs of respiratory disease reached the terminal stage and was euthanised one day before the end of the experiment. This also occurred in the case of one chicken in group 2, which received the traditional dosage. Another individual in this group succumbed to the disease one day before the conclusion of the experiment. Group 3, which was experimentally infected but untreated, showed 100% mortality within the first 24 h from the start of the experiment; two chickens died within 8 h after infection, while the other eight chickens died the day after infection. In control group 4, which was uninfected and untreated, all chickens survived to the end of the experiment.

Necropsies were performed on all deceased individuals, and samples of the internal organs (liver, kidneys, lungs, air sac swabs) were taken for bacteriological examination. All these samples were cultured for the presence of *E. coli*. Further typing confirmed that the strain isolated from the samples was the strain used for the experimental infection.

Survival curves were generated for each group based on the mortality counts following experimental *E. coli* infection utilising the Kaplan–Meier survival function (Figure 4). Statistical analysis using the log-rank test revealed a significant difference (*p*-value < 0.0001) in survival between the groups treated with the experimental and traditional dosage compared to the untreated experimentally infected group. However, there was no statistically significant difference (*p*-value > 0.05) between the experimental vs. traditional drug-treated groups.

## 3. Discussion

Antimicrobials are still the main tool for combatting the emergence or mitigating the course of infection in poultry colibacillosis [15,16]. In poultry farms, the commonly used antimicrobial to solve this problem is trimethoprim/sulfamethoxazole combination, which is a potent ‘broad spectrum’ antimicrobial agent [17]. This synergic combination effectively targets a wide range of bacterial infections, making it a versatile and valuable therapeutic option for infectious diseases. In vitro, it is active against a wide range of organisms, including Gram-positive and Gram-negative aerobic bacteria, chlamydia, *Nocardia* (actinomycetes), some mycobacteria and protozoa, and many anaerobic bacteria [13,18,19]. However, the use of ‘broad spectrum‘ antibiotics for treatment also creates a selection pressure for the emergence of resistant causative agents, which can contrary lead to treatment failure and increased economic losses for farmers [16,20,21].

The effectiveness of antibiotic treatment is based on two variable factors—the duration of treatment and the dose. These two factors are usually recommended by the manufacturer of each drug as a traditional treatment with fixed treatment doses administered for a given period of time. These traditional treatments are used routinely in both human and veterinary medicine [22]. Although traditional treatments can be effective, they may not be optimal in all cases and there is an increasing emphasis on the need to incorporate the results of pharmacokinetic and pharmacodynamic analyses into treatment planning [22,23]. Studies that have aimed to compare the efficacy of traditional and lower therapeutic doses (however, the first therapeutic dose is usually recommended higher) described that precisely defined lower doses of antibiotics, based on PK and PD studies, are equally clinically effective and can minimise drug side effects and are an appropriate strategy for reducing the risk of antibiotic resistance in bacterial populations [22,24].

Given the contemporary challenge of antibiotic resistance, our study recommends targeted interventions using modern techniques that are crucial for effectively combatting this threat and aim to minimise the dose ensuring effective treatment. The use of antimicrobials is currently approached within the framework of the World Health Organization’s (WHO) “One Health” concept, which aims to ensure the balance and optimisation of human, animal, and environmental health [25]. Legislation, both at the national and international levels, aims to counter bacterial resistance through measures that promote the prudent use of existing antimicrobials, with a focus on targeted treatments based on accurate pathogen identification and susceptibility assessments. At the same time, antimicrobials are retained primarily for the treatment of bacterial infections in humans; therefore, their use in veterinary medicine is restricted or, in some cases, prohibited and needs to be monitored [26]. These principles are defined in the EU Regulation 2019/6 of the European Parliament and of the Council on Veterinary Medicinal Products, which also requires the EU Member States to collect data on the consumption of prescribed antibiotics and their resistance [9]. Our investigation aligns with the legislative principles and contributes insights into the pharmacokinetics, antimicrobial efficacy, and therapeutic potential of the trimethoprim and sulfamethoxazole combination in a chicken model infected with *E. coli*.

Even though combinations of sulfonamides and trimethoprim are still widely used in veterinary practice against serious infectious diseases of bacterial origin, especially respiratory and infectious diseases, in various animal species, adequate information for their application in poultry is lacking [27]. Furthermore, the PK studies performed on different animal species for various sulfonamides/trimethoprim combinations were mostly published more than 20 years ago and conducted using analytical techniques available at that time, most commonly liquid chromatography with UV detectors. For these reasons, we used a highly sensitive analytical method based on UHPLC with high-resolution mass spectrometry to determine plasma concentrations for the accurate identification of the targeted analytes. Thus, our study is the first to determine PK curves and parameters for the single and repeated oral administration of sulfamethoxazole potentiated with trimethoprim with a recommended dosage for poultry using modern analytical techniques.

In the literature considering a similar subject, trimethoprim 5.5 mg combined with sulfamethoxazole 27.5 mg per 1 kg b.w. was administered through drinking water in ROSS line broilers during early fattening [12]. Both sulfamethoxazole and trimethoprim showed rapid absorption while having better interstitial or tissue distribution, respectively. Variations in C_max_ were observed, with trimethoprim ranging from 1.3 to 2.1 µg mL^−1^ and sulfamethoxazole from 47 to 59 µg mL^−1^. In a comparable PK study conducted on hens after oral dose application, published in 1985, the C_max_ in the plasma was C_max_ = 54.5 μg mL^−1^ for sulfamethoxazole and C_max_ = 1.2 μg mL^−1^ for trimethoprim [28]. In this study, however, an application dose approximately three times higher than the one employed in our study was used, i.e., a dose of 96 mg (80 mg sulfamethoxazole + 16 mg trimethoprim) per 1 kg b.w. The range of variability on the PK curve (Figure 1), especially around the time of T_max_, was determined by the design of the animal experiment, which was different from the design of PK studies in human medicine. One animal had to be sacrificed each time to collect a sufficient amount of one plasma sample for analysis. Thus, each single plasma sample came from a different animal each time, because 19-day-old broilers were too small to provide sufficient plasma samples repeatedly. One of the objectives of the work was to focus the therapy on broilers in the early phase of fattening. The individual biological variability of each animal increased the general variability of the treatment intervention.

In our study, the validated analytical method demonstrated the precision and sensitivity necessary for quantifying both drugs in chicken sera, setting the stage for robust PK analyses. The main tool for predicting steady-state drug concentration after oral administration and determining the effective dose for the target broiler animal was the use of clinically applied PK measurements, and the second was in vitro antimicrobial efficacy assessment. A pharmacodynamic parameter (concentration or time dependence) that is primarily related to bacterial killing has not been clearly established to date. Sulfamethoxazole/trimethoprim combination has a variable bacteriostatic and bactericidal effect during its course of action. In general, sulfamethoxazole and trimethoprim have bacteriostatic activity that is time-dependent, with the potential for concentration-dependent bactericidal activity for sensitive organisms [29,30]. Against Gram-positive organisms such as *S. aureus*, the sulfamethoxazole/trimethoprim has bactericidal activity, as reported in an earlier study [31]. Conversely, for Gram-negative organisms, it has been hypothesised that sulfamethoxazole/trimethoprim has concentration-dependent killing of *E. coli*, but clear evidence has not been reported [29].

Since the pharmacodynamics are still unclear, we primarily based our design on concentration dependence, and by comparing the C_max_ with the MIC of each active ingredient, an effective experimental dose of a 4:1 fixed combination for repeated dosing of 20 mg (combined with 16 mg sulfamethoxazole + 4 mg trimethoprim) per 1 kg live weight of broilers was designed. We set a predictive value of C_max_ 1.5 times higher than MIC ranging from ≤0.5/9.5 mg L^−1^ to 1/19 mg L^−1^ for susceptible isolates, i.e., C_max_ = 1.5/28.5 mg L^−1^. Subsequently, we calculated the volume of distribution Vd for both substances using a Monte Carlo simulation method. For the simulation, we used the corresponding dose D and C_max_ ± SD from the PK study. From the predicted C_max_ and simulated Vd range, we backcalculated the new dosage. However, to estimate the optimum dosage, we also worked secondarily with time dependence and AUC—MIC ratio. Using these experimental approaches, it was possible to optimise and individualise the dosage under different conditions (different doses, single or repeated applications), which was further experimentally verified in a clinical study.

Using these experimental approaches, it was possible to optimise and individualise the dosing under different conditions (different doses, single or repeated applications). By comparing the C_max_, T_max_, and AUC against the microbiological efficacy of each active ingredient, an effective experimental dose of a fixed 4:1 combination for repeated dosing was proposed as 20 mg (in combination with 16 mg sulfamethoxazole + 4 mg trimethoprim) per 1 kg b.w. of broilers. Our antimicrobial efficacy assessment, based on MICs, highlighted the potency of the trimethoprim and sulfamethoxazole combination against *E. coli* isolates. The observed susceptibility and resistance categories provide valuable information for guiding dosage recommendations and optimising treatment strategies in the context of bacterial infections. The results of MICs for trimethoprim/sulfamethoxazole in this study can be compared with the results of MICs for *E. coli* isolates originating from poultry in the National Antibiotic Program monitoring from 2017 to 2021 published by the State Veterinary Administration of the Czech Republic. In all years of national monitoring, MIC_50_ and MIC_90_ were found to be ≤0.25/4.75 mg L^−1^ and >32/608 mg L^−1^, respectively. The MIC_50_ and the MIC_90_ were detected as limited values of monitored concentration. The percentage of susceptible strains in our study was 79.6%, which is comparable to the national monitoring results (80.7% in 2017; 81.3% in 2018; 82.1% in 2019; 81.6% in 2020; and 84.3% in 2021) [32]. The results of MIC monitoring of isolates in the Czech Republic may not fully correspond to results from other countries. Therefore, it is always necessary to emphasise that before starting any antibiotic treatment, it is necessary to perform antibiotic susceptibility testing of the bacterial pathogen.

Based on the proposed dosing determined by PK parameters, the efficacy of the newly proposed dosage of these substances in an altered 4:1 ratio was verified in broilers experimentally infected with *E. coli*. The notable difference in survival outcomes between the groups treated with experimental and traditional dosing compared to the untreated group reinforces the potential of the drug combination in mitigating the adverse effects of *E. coli* infection. The results of this experiment demonstrated the statistically significant efficacy of the experimental treatment when compared to the untreated group. The potency of the experimental treatment was comparable to that of the conventional treatment, as demonstrated by the survival curves in Figure 4. Furthermore, the experimental treatment of broilers in the early fattening stage did not adversely affect their weight gain, average daily gain or feed conversion. Statistically, the *t*-test showed no significant difference (*p*-value = 0.156) in live weight and AGD (*p*-value = 0.239) between group 3 (experimentally infected and untreated) and group 1 (experimentally treated) at the beginning and end of the experiment. The changes in broiler weight during the experiment are shown in Figure 2 and Figure 3. Even though the group treated with the experimentally proposed dosing ratio is a relatively small sample for extensive statistical evaluation, the probability of survival was slightly higher (10%) in the group treated with the experimentally proposed dosing ratio than in the group treated with the traditional dosing ratio. The chickens in groups 1 and 2 were treated at three per os doses at 24 h intervals as proposed by the EMA [33], where the recommended treatment period for broilers, which is usually 3 to 4 days for registered poultry veterinary medicines containing trimethoprim and sulfamethoxazole. This is also described in [34], where the recommended treatment period is usually 3 days, i.e., until the mortality in breeding ceases and the animals do not show clinical symptoms. Moreover, the authors describe that, although the mortality of individuals may continue to increase at the beginning of treatment, these are probably very sick individuals who are not able to receive a sufficient dose of the drug necessary for effective treatment.

Our study thus suggests implementing a lower dose of sulfamethoxazole/trimethoprim in the ratio of 4:1 into the protocols for the treatment of poultry colibacillosis. Even if the duration of treatment would then be extended to achieve greater treatment success and the survival of all the animals, the reduced dose could still help to prevent the development of antibiotic resistance. Another possibility is to shorten the drug administration intervals to, for example, 12 h while maintaining a reduced dose of 20 mg per 1 kg b.w. of broilers. Both of these proposed approaches are the subject of our future experiments, when we plan to verify the effectiveness of both in field conditions in poultry farms on a larger number of individuals.

In summary, this study offers a comprehensive approach to poultry colibacillosis treatment, incorporating modern analytical techniques and practical considerations for optimal dosing, with the potential to significantly reduce the use of antimicrobials in veterinary medicine. This reduction is fundamental in the ongoing efforts to combat bacterial infections in poultry while navigating the challenges posed by antibiotic resistance and the need for sustainable veterinary practices. Moreover, this substantial reduction in the effective dose of antimicrobials has a multitude of positive implications for veterinary medicine. It not only ensures the effective protection of domestic chickens with minimal stress on their organs but also shortens the withholding period in the food chain. Moreover, this reduction in the effective dose contributes to mitigating the residual burden on the environment by limiting the impact of residues of drugs and their metabolites from the animals on farms and their surroundings, particularly in wastewater. Finally, from an economic viewpoint, our findings open up a promising avenue for cost-effective poultry health management. The optimisation of antimicrobial dosing not only minimises the expenses associated with treatment but also aligns with sustainable practices, reducing economic losses for farmers in the long term.

## 4. Materials and Methods

### 4.1. Chemicals for Analytical Methods

The analytical reference standards of trimethoprim (EurPh reference standard impurity B; CAS n.: 738-70-5) and sulfamethoxazole (EurPh reference standard impurity F; CAS n.: 723-46-6), including the analytical internal standards of trimethoprim-d_9_ and sulfamethoxazole-^13^C_6_ (Vetranal^TM^), were purchased from Merck (Darmstadt, Germany), together with the acetonitrile hypergrade for LC-MS (LiChrosolv) and formic acid. The deionised water was prepared with a water treatment system from Goldman Water Ltd. (Zalelenc, Czech Republic). The calibration solutions, Pierce ESI, were purchased from Thermo Fisher Scientific (Waltham, MA, USA).

### 4.2. Sample Preparation and Analytical Method

Samples were prepared, and the analytical method was used according to a previously published study [12]. In brief, 1 mL of serum was spiked with internal standards to achieve a concentration of 100 ng mL^−1^. Then, 200 µL of the spiked serum was mixed with 200 µL of acetonitrile, vortexed for 3 min, and centrifuged (20 min, 14,100× *g*). A total of 360 µL of supernatant was evaporated to dryness under a stream of nitrogen (t = 30 °C), and the formed pellet was dissolved in 150 µL of 30% acetonitrile in water. The resulting solution was transferred to a chromatographic vial, from which 5 µL was taken and injected into the measuring system.

The kinetics of the time-dependent changes in the broiler plasma concentrations of the target analytes, trimethoprim and sulfamethoxazole, were measured with analytical equipment using highly sensitive and accurate methods based on liquid chromatography–mass spectrometry (LC-MS/MS). The LC system was a Vanquish Horizon (Thermo Fisher Scientific, Waltham, MA, USA) equipped with a binary pump with vacuum degassing and an autosampler. The high-resolution mass spectrometer was a Q Exactive (Thermo Fisher Scientific, Waltham, MA, USA) equipped with a heated electrospray ion source measured in positive mode (H-ESI+). An analytical column, a Luna Omega 1.6 µm Polar C18 100 mm × 2.1 mm (Phenomenex, Torrance, CA, USA), was used for the chromatographic separation. The analytical methods, including the HPLC gradient conditions and composition of the acetonitrile mobile phases, have been described in detail in a previous study, together with the mass/charge of the precursor and product ions [12].

### 4.3. Method Validation and Pharmacokinetic

The analytical method used to determine the PK of the target drugs was validated in the range of specificity (identification), precision, accuracy, linearity, linearity range, limit of detection (LOD), and limit of quantification (LOQ) according to the recommendations of the EMA (Guidelines VICH GL 2 and VICH GL 49) [35,36]. The specification and identification of the target analytes in the matrices used were based on chromatography (retention time; RT ± 10%) and characteristic mass spectra measured at a high resolution (mass accuracy; MA < 5 ppm). Accuracy was determined based on recovery, and precision was determined based on repeated measurements of the model serum samples fortified with the target analytes at three different concentration levels. The intermediate method’s precision was determined through repeated measurements at three different times (days). Linearity was verified by measuring a matrix calibration curve at six concentration levels (including a blank), with six repeated measurements for each concentration. The calibration range was from 0 to 50 μg mL^−1^ for sulfamethoxazole and from 0 to 2 μg mL^−1^ for trimethoprim. The sensitivity of the method, expressed as the LOD and LOQ, was determined based on the standard deviation of the blank response (SDy) and the slope of the matrix calibration curve.

The kinetics of the changes in the plasma concentrations of trimethoprim and sulfamethoxazole over time, after single and repeated administration to the broilers, were characterised using a single-compartment open pharmacological model. Graphical techniques were used to estimate the pharmacokinetic profiles of trimethoprim and sulfamethoxazole, which included plotting the concentration (mean ± SD) versus time.

Key pharmacokinetic parameters, including the C_max_, T_max_, area under the curve up to the last sampling (AUC_t_), and area under the curve extrapolated to infinity (AUC_∞_), were calculated directly from the PK curves based on the trapezoidal rule, in accordance with the guidance in the EU Directive (Guideline EMEA/CVMP/EWP/133/1999) [37].

### 4.4. Antimicrobial Efficacy of Trimethoprim with Sulfamethoxazole

The antimicrobial efficacy of trimethoprim with sulfamethoxazole was assessed by determining the minimal inhibition concentrations (MICs) via the microdilution method using 113 field isolates of APEC, according to EUCAST [13] and the Clinical Laboratory Standards Institute (CLSI) guidelines [38], in a ratio of 1:19 trimethoprim and sulfamethoxazole. The sets for MIC determination were manufactured at the Veterinary Research Institute in Brno, Czech Republic, in accordance with the CLSI and EUCAST methodologies [13,39]. The MIC_50_ and MIC_90_ values were determined from cumulative results regarding the lowest antimicrobial concentration in mg/L that inhibits the growth of 50% and 90% of isolates [40].

The isolates came from samples collected from sick or dead chickens from 17 broiler farms in the Czech Republic between 2019 and 2022. The isolates that originated from the samples from the same farm were included in the testing of susceptibility/resistance to trimethoprim and sulfamethoxazole combination at least once every 6 months. The methodology of sample collection is based on the methodology of the Czech National Antibiotic Program defined by the State Veterinary Administration of the Czech Republic [32]. Isolates from the same farm were included in the MIC monitoring results in this study only if they were identified to have a different *E. coli* pathotype as determined based on detecting of the presence of individual virulence genes. Details of the diagnosis of *E. coli* isolates and the determination of pathotypes have been described in a recently published study [41].

### 4.5. Animal Experiment

The animal experiment was carried out in the accredited animal facilities of the Veterinary Research Institute in Brno after being approved by the Branch Commission for Animal Welfare of the Ministry of Agriculture of the Czech Republic (permission MZe No. 2071). The animal care protocol for this experiment followed the Czech guidelines for animal experimentation.

A total of forty one-day-old broilers of both sexes representing the ROSS line were included in the experiment; the sex ratio did not affect the study. The broilers were divided into four groups of ten (Table 2).

Groups 1, 2, and 3 were infected at the age of 19 days with a strain associated with APEC, while group 4 was not infected. Group 1 was treated with an experimental dosage, a combination of trimethoprim and sulfamethoxazole, in a 1:4 weight ratio at a dose of 20 mg kg^−1^ of live weight; this dosing was derived from the results of the analytical determination of the PK properties of both substances. The experimental dose was derived based on the predictive C_max_ and simulated distribution volume Vd for both drugs. Estimation of Vd was performed by Monte Carlo simulation from data obtained in the PK study. The predictive C_max_ was determined as 1.5 times the MIC breakpoint. Group 2 was treated with a traditional dosage, a combination of trimethoprim and sulfamethoxazole, in a 1:5 weight ratio at a dose of 33 mg kg^−1^ of live weight, as recommended by the manufacturers of the commercial drugs. Groups 3 and 4 were not treated. The broilers had unrestricted access to feed and drinking water throughout the experiment. The housing conditions (temperature, humidity, light regime) followed the Technological Procedures for breeding broilers of the ROSS line [42].

The infectious *E. coli* strain (CAPM 6685) originated from a yolk sac infection, namely, ST95, serotype O1-NT, phylogenetic group B2, virulence genes *ompT*, *iroN*, *iss*, *iutA*, and *hlyF*, which are associated with APEC strains [18,43,44,45], without the presence of antibiotic resistance genes. The *E. coli* strain is collected in the Collection of Animal Pathogenic Microorganisms at the Veterinary Research Institute (Brno, Czech Republic, https://capm.vri.cz (accessed on 30 November 2023). The strain was identified by MALDI-TOF mass spectrometry, antibiotic resistance assays were performed (determination of minimal inhibition concentration using a microdilution method), and the strains were subjected to whole-genome sequencing using the Illumina NextSeq and MiSeq platforms. The data were annotated and compared with the ResFInder 3.1, PlasmidFinder 2.0, SeroTypeFinder 2.0, and MLST 2.1 databases (Centre for Genomic Epidemiology, Technical University of Denmark; http://www.genomicepidemiology.org/ (accessed on 30 November 2023). CARD (Comprehensive Antibiotic Resistance Database) and VFDB (Virulence Factor Database) and a phylogenetic group were identified (http://clermontyping.iame-research.center/ (accessed on 30 November 2023). Then, 0.2 mL of a bacterial culture of the infectious strain with an adjusted density of 5 × 10^9^ CFU mL^−1^, previously incubated for 16 h at 37 °C in a brain–heart infusion (Thermo Fisher Scientific CZ, Praha, Czech Republic), was injected into the left air sac of all the broilers in groups 1–3. The applied experimental infection model was based on previous experiments within the approved protocol of animal experimentation (permission MZe No. 2071).

The time schedule of the experiment is given in Table 3. Immediately after infection, the treatment was started. The chickens in groups 1 and 2 were treated with the above-described drugs (experimental dosage, group 1; traditional dosage, group 2) at three doses of 1 mL, administered individually per os with a probe at 24 h intervals. The health status of each chicken was monitored regularly. The weight of the live chickens in all the groups was recorded from the day of infection until the end of the experiment. Chickens in the terminal stage of the disease were euthanised. The terminal stage of the disease was defined as a stage that cannot be cured or adequately treated and is expected to result in death—chickens unable to move, unable to drink and eat, with very high fever, breathing heavily without responding to external stimuli. Seven days after infection, the experiment was terminated by culling all the surviving chickens. Necropsies were performed on all the deceased and euthanised chickens.

### 4.6. Statistical Analysis

The measured data were statistically evaluated using Statistica software (TIBCO Statistica^®^ 13.3.0). The pharmacokinetic curves and PK parameters were evaluated using GraphPad Prism v. 8.4.3 (GraphPad Software, Boston, MA, USA). All the results are reported as the mean ± standard deviation (SD) or as a 95% confidence interval (CI). The *t*-test was used to compare the mean body weights of the broilers at the beginning and end of the experiment. The log-rank test was used to test for statistically significant differences in survival between the treated and untreated groups experimentally infected with *E. coli*. A calculated *p*-value less than 0.05 (*p*-value < 0.05) was considered statistically significant.

## 5. Conclusions

Based on the results of the bioassay performed on the target broiler chickens, it can be concluded that the experimental 1:4 ratio of trimethoprim and sulfamethoxazole combination has an efficacy at least similar to that of the traditional 1:5 ratio based on a similar composition. It has also been shown that a much lower dose of the experimental dosage (20 mg kg^−1^ of live weight), compared to the one previously recommended (45 mg kg^−1^ of live weight), is effective for the treatment of broiler chickens. Given the results of this study, the authors recommend incorporating the experimental results into the development of effective treatment protocols for *E. coli* infections in chickens together to gradually reduce antibiotic resistance. The authors recommend that the drug be administered in four or five doses for all animals in the treated groups or in shortened time intervals in order for them to survive the infection.

By minimising the selection pressure typically associated with antibiotic treatments, this research contributes to broader efforts to preserve the efficacy of antimicrobials in the face of growing resistance challenges. However, the study emphasises the need for ongoing research to monitor resistance patterns, optimise dosages, and ensure the responsible use of antimicrobials in alignment with regulatory frameworks. The multifaceted nature of trimethoprim and sulfamethoxazole combination positions it as a valuable asset in poultry health, warranting further exploration and careful consideration in the broader context of antimicrobial stewardship and the “One Health” strategy.

## Figures and Tables

**Figure 1 antibiotics-13-00011-f001:**
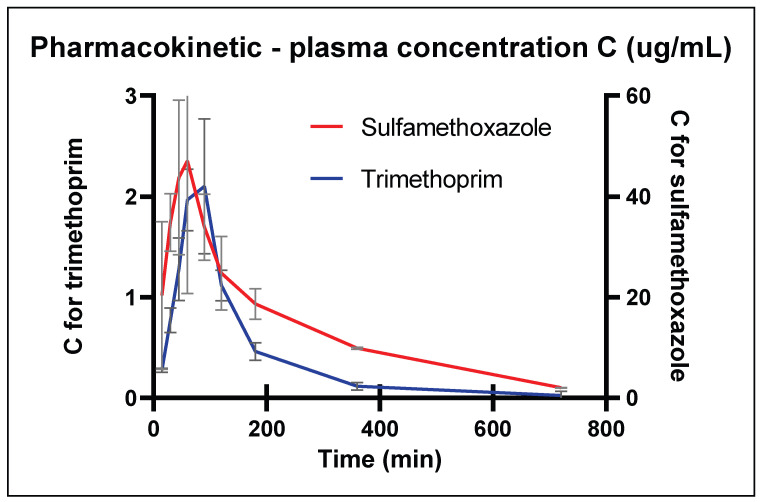
The pharmacokinetic curve showing the dependence of plasma concentration on sampling time points (mean ± SD). Y axis on the left—trimethoprim concentration in the range of 0 to 3 μg mL^−1^; Y axis on the right—sulfamethoxazole concentration in the range of 0 to 60 μg mL^−1^.

**Figure 2 antibiotics-13-00011-f002:**
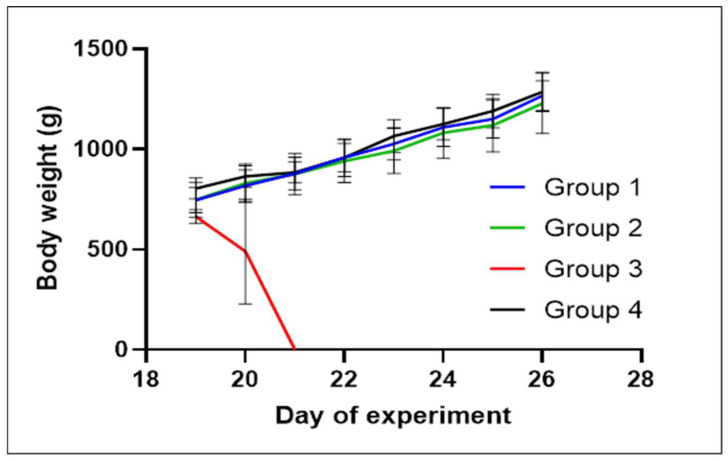
Weight and survival of chicks in experimentally infected groups 1 (experimental dosage), 2 (traditional dosage) and 3 (no treatment) versus control group 4 (no infection) were monitored during the experiment. Points on the curves represent the mean ± SD with n = 10 in each group.

**Figure 3 antibiotics-13-00011-f003:**
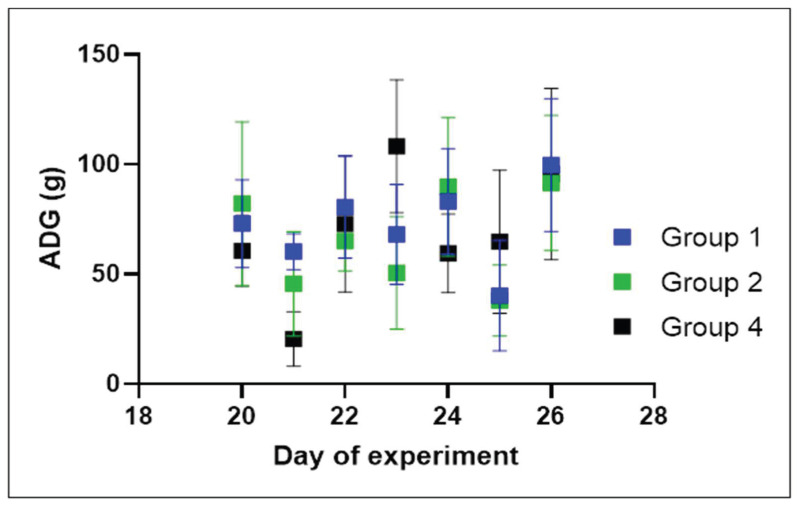
The average daily gain (ADG) of chicks in experimentally infected groups 1 (experimental dosage) and 2 (traditional dosage) versus control group 4 (no infection) was monitored during the experiment. Box and whiskers represent mean ± SD with n = 10 in each group.

**Figure 4 antibiotics-13-00011-f004:**
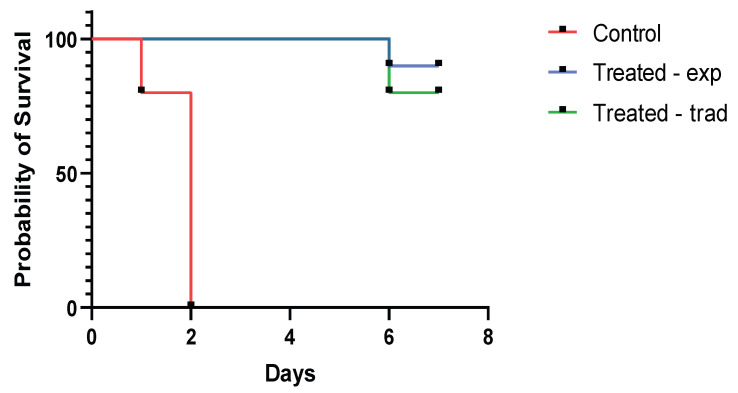
Kaplan–Meier survival curves following trimethoprim and sulfamethoxazole treatment with traditional and experimental dosing after experimental *E. coli* infection compared to untreated infected group.

**Table 1 antibiotics-13-00011-t001:** Distribution of MICs in APEC isolates (n = 113).

Trimethoprim/Sulfamethoxazole Concentrations (mg L^−1^)	Number of Isolates	%
>64/1216	20	17.7
64/1216	2	1.8
32/608	0	0
16/304	0	0
8/152	0	0
4/76	0	0
2/38	1	0.9
1/19	3	2.6
≤0.5/9.5	87	77.0

**Table 2 antibiotics-13-00011-t002:** Division of broilers into groups.

Group	Experimentally Infected	Treatment by the Experimental Dosage	Treatment by the Traditional Dosage
1	yes	yes	no
2	yes	no	yes
3	yes	no	no
4	no	no	no

**Table 3 antibiotics-13-00011-t003:** Time schedule of the experiment.

Day	Activities with Animals
1	Intake of day-old chicks, divided into 4 groups of n = 10
19	Weighing of chickens, experimental infection (groups 1, 2, and 3), initiation of treatment by group (groups 1 and 2)
20	Continuation of treatment by group (groups 1 and 2), weighing of chickens
21	Continuation of treatment by group (groups 1 and 2), weighing of chickens
22	Weighing of chickens
23	Weighing of chickens
26	End of experiment

## Data Availability

Raw data supporting the conclusions of this study are available from the authors upon request.

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
