# Peer review of "Dosage Optimisation of Trimethoprim and Sulfamethoxazole for the Treatment of an Avian Pathogenic Strain of Escherichia coli in Broiler Chickens"

_antibiotics, 2023, doi:10.3390/antibiotics13010011_

Round 1
Reviewer 1 Report
Comments and Suggestions for Authors
1. I offer + 1 keyword "E. coli"
2. Line 103-109. Write each aim on a separate line.
3. Line 124-128 sounds like the last paragraph of the "results" section, not the "introduction".
4. Table 1. The sum of strains in the second column is not 114 as written in the title, but 113, check it.
5. I think tables 2-3-4-5 should be combined and display the average value (+-error) for each group, rather than showing each chicken separately. Or give these tables in the supplementary material.
6. In figure 6, it is worth noting whether there is a significant difference (p < or > 0.05) between experimental and traditional dosing.
7. Lines 383 and 386. I would not use the terms "experimental drug" and "traditional drug", because we are not talking about "drug", but about "dosing" of already existing known drugs. Therefore, it may be worth using the terms "experimental dosage" / "traditional dosage" here and in the manuscript as a whole.
8. The title of the manuscript should be corrected. The manuscript is not about a "novel pharmaceutical compound", it is about a new proposed dosage of existing compounds. As an option, the title of the manuscript "Optimization of the dosage of trimethoprim and sulfamethoxazole for the treatment of an avian pathogenic strain of Escherichia coli in broiler chickens" is possible.
9. In general, the manuscript is interesting and useful. It is possible in the "discussion" to provide information about other cases when reducing the use of antibiotics or reducing the dose did not lead to worsening of treatment or to an increase in the number of infectious processes. For example, among humans, increased control over antibiotics does not lead to an increase in the number of cases of disease. It is also possible in the "discussion" to talk about the reasons why lower doses of antibiotics are not used, for example because of fear.
Author Response
Response to Reviewer 1 Comments:
Thank you very much for taking the time to review this manuscript and your prompt response!
- I offer + 1 keyword "E. coli"
- The keyword E. cli has been added.
- Line 103-109. Write each aim on a separate line.
- Each aim has been written on a separate line.
- Line 124-128 sounds like the last paragraph of the "results" section, not the "introduction".
- Lines 124 – 128 have been moved to the „results“ section.
- Table 1. The sum of strains in the second column is not 114 as written in the title, but 113, check it.
- The number of strains in the Table 1 title and in the „Material and Methods“ section has been corrected, the data in the „%“ column have been adjusted accordingly with it.
- I think tables 2-3-4-5 should be combined and display the average value (+-error) for each group, rather than showing each chicken separately. Or give these tables in the supplementary material.
- Yes, we agree. Figures 2, 3, 4 and 5 have been integrated and revised. The data will be shown in supplementary material.
- In figure 6, it is worth noting whether there is a significant difference (p < or > 0.05) between experimental and traditional dosing.
- Yes, we agree. Statistical significance has been added to the text for Figure 6 (new Fig.4) to compare Group 1 vs 2.
- Lines 383 and 386. I would not use the terms "experimental drug" and "traditional drug", because we are not talking about "drug", but about "dosing" of already existing known drugs. Therefore, it may be worth using the terms "experimental dosage" / "traditional dosage" here and in the manuscript as a whole.
- The terms „experimental drug“ and „traditional drug“ have been modified to „experimental dosage“ and „traditional dosage“ thorough the manuscript according to your recommendation.
- The title of the manuscript should be corrected. The manuscript is not about a "novel pharmaceutical compound", it is about a new proposed dosage of existing compounds. As an option, the title of the manuscript "Optimization of the dosage of trimethoprim and sulfamethoxazole for the treatment of an avian pathogenic strain of Escherichia coli in broiler chickens" is possible.
- The title of the manuscript has been modified according to your recommendation.
- In general, the manuscript is interesting and useful. It is possible in the "discussion" to provide information about other cases when reducing the use of antibiotics or reducing the dose did not lead to worsening of treatment or to an increase in the number of infectious processes. For example, among humans, increased control over antibiotics does not lead to an increase in the number of cases of disease. It is also possible in the "discussion" to talk about the reasons why lower doses of antibiotics are not used, for example because of fear.
- The „Discussion“ section has been modified and further information has been added.
Thank you for all yours comment.
Please note that the manuscript was also edited based on comments from three other reviewers, so the revised version has more changes than you requested.
Reviewer 2 Report
Comments and Suggestions for Authors
This study proposes a novel combination containing similar active substances in a newly established ratio of 1:4, with a recommended dosage of 20 mg kg-1 of live weight administered at 24-hour intervals for 3 to 5 days. Overall, the proposed new dosage of trimethoprim-sulfamethoxazole for broiler chickens has the potential to reduce the use of antimicrobials and promote their prudent application. However, I believe further research is needed to address the potential drawbacks of this reduced dosage before it can be widely adopted.
Major comments:
The study would be strengthened by the addition of a group of chickens that were infected with E. coli but not treated with any drugs. This would allow the authors to compare the efficacy of the experimental and traditional drugs to an untreated control group.
The new formulation requires further testing to evaluate its long-term safety and efficacy in a larger population of broiler chickens. I do have some concerns about the animal experiment. Specifically, I am concerned about the number of chickens that were used in the study.
The potential impact of the new formulation on the development of antimicrobial resistance needs to be carefully assessed in a larger population.
The authors should provide more information about the specific criteria used to define the terminal stage of the disease. This would make it easier for other researchers to interpret the results of the study.
The authors should provide more information about the specific criteria used to define the cutoff values for the LOD and LOQ an also about the specific methods used to validate the pharmacokinetic model.
The manuscript is well written, but the title and introduction are both too long and contain unnecessary information.
The authors could provide more information about the strain of APEC that was used in the antimicrobial efficacy study and how they tested for it in some more detail.
Despite these comments, the study provides valuable information on the pharmacokinetics and antimicrobial efficacy of this combination of trimethoprim and sulfamethoxazole for the treatment of colibacillosis in broiler chickens.
Author Response
Response to Reviewer 2 Comments:
Thank you very much for taking the time to review this manuscript and your prompt response!
This study proposes a novel combination containing similar active substances in a newly established ratio of 1:4, with a recommended dosage of 20 mg kg-1 of live weight administered at 24-hour intervals for 3 to 5 days. Overall, the proposed new dosage of trimethoprim-sulfamethoxazole for broiler chickens has the potential to reduce the use of antimicrobials and promote their prudent application. However, I believe further research is needed to address the potential drawbacks of this reduced dosage before it can be widely adopted.
- We strongly agree that further large-scale experiments need to be conducted to generalize our results. The manuscript provides results obtained within the scope of the research project that we find interesting and hope will be supported by further research.
Major comments:
The study would be strengthened by the addition of a group of chickens that were infected with E. coli but not treated with any drugs. This would allow the authors to compare the efficacy of the experimental and traditional drugs to an untreated control group.
- We do not really understand the comment - a group of chickens that were infected E. coli but not treated with any drug was included in the experiment (group 3). This information is in Table 2. However, the second line in Table 3 has been modified based on your comment. Or did we misunderstand this comment?
The new formulation requires further testing to evaluate its long-term safety and efficacy in a larger population of broiler chickens. I do have some concerns about the animal experiment. Specifically, I am concerned about the number of chickens that were used in the study.
- Yes, we know, the new formulation reguires further testing, laboratory analysis and validation of results with larger animal population. The number of chickens in each group in our experiment was based on the financial possibilities of the grant and was chosen to allow its basic statistical evaluation. As mentioned above, we hope that there will be an opportunity for further research on this issue.
The potential impact of the new formulation on the development of antimicrobial resistance needs to be carefully assessed in a larger population.
- The answer to this comment is related to the previous answer. The impact on the development of antimicrobial resistance needs to be further tested in follow-up studies with much larger numbers of animals over a longer period of time.
The authors should provide more information about the specific criteria used to define the terminal stage of the disease. This would make it easier for other researchers to interpret the results of the study.
- The criteria and definition of the terminal stage of the disease were described in „Material and Method“ section.
The authors should provide more information about the specific criteria used to define the cutoff values for the LOD and LOQ an also about the specific methods used to validate the pharmacokinetic model.
- Yes, we agree. The authors added to the manuscript a link to their previous work where this information is presented in detail.
The manuscript is well written, but the title and introduction are both too long and contain unnecessary information.
- The title of the manuscript has been corrected and the „Introduction“ section have been shortened.
The authors could provide more information about the strain of APEC that was used in the antimicrobial efficacy study and how they tested for it in some more detail.
- More details about the infection strains and its testing have been added to „Material and Method“ section.
Despite these comments, the study provides valuable information on the pharmacokinetics and antimicrobial efficacy of this combination of trimethoprim and sulfamethoxazole for the treatment of colibacillosis in broiler chickens.
Thank you for all yours comment.
Please note that the manuscript was also edited based on comments from three other reviewers, so the revised version has more changes than you requested.
Reviewer 3 Report
Comments and Suggestions for Authors
The author tested a new combination of drugs to treat colibacillosis caused by E. coli infection in broiler chickens. Compared to the traditional drug combinations, the new combination has the similar efficiency but less dosage which will not only save the cost but also reduce the risk of side effect. This is the good point of this paper.
However, the animal experiment design can be improved. I have two suggestions.
1. The author compared the weight of individuals in each group, I recommend combining four group data together to show if there are any differences among different groups. Especially the new combination group and the traditional group.
2. For the drug test, the author has to add two more groups: the uninfected chickens with traditional drug treatment, and the uninfected chickens with new combination drug treatment. Please record the weight and survival curves of these two groups. These two groups can help to compare the toxicity of the new combinations and traditional drugs.
Thank you!
Author Response
Response to Reviewer 3 Comments:
Thank you very much for taking the time to review this manuscript and your prompt response!
The author tested a new combination of drugs to treat colibacillosis caused by E. coli infection in broiler chickens. Compared to the traditional drug combinations, the new combination has the similar efficiency but less dosage which will not only save the cost but also reduce the risk of side effect. This is the good point of this paper.
However, the animal experiment design can be improved. I have two suggestions.
- The author compared the weight of individuals in each group, I recommend combining four group data together to show if there are any differences among different groups. Especially the new combination group and the traditional group.
- Yes, we agree. Figures 2, 3, 4 and 5 have been integrated and revised. The data will be shown in supplementary material.
- For the drug test, the author has to add two more groups: the uninfected chickens with traditional drug treatment, and the uninfected chickens with new combination drug treatment. Please record the weight and survival curves of these two groups. These two groups can help to compare the toxicity of the new combinations and traditional drugs.
- Yes, we strongly agree that further large-scale experiments need to be conducted to generalize our results. The manuscript provides results obtained within the scope of the research project that we find interesting and hope will be supported by further research. The design of the animal experiment was based on the financial possibilities of the grant. Unfortunately, conducting a new experiment with the other two groups suggested in the manuscript improvement comment cannot be done as part of the revision process due to time and financial limitations.We hope that there will be an opportunity for further research on this issue.
Thank you for all yours comment.
Please note that the manuscript was also edited based on comments from three other reviewers, so the revised version has more changes than you requested.
Reviewer 4 Report
Comments and Suggestions for Authors
I appreciate the opportunity to review the manuscript titled "Managing experimental infection in broiler chickens caused by an avian pathogenic strain of Escherichia coli using a novel pharmaceutical compound containing trimethoprim and sulfamethoxazole" for potential publication in Antibiotic. While the manuscript is interesting, I have significant concerns that need to be addressed in order to consider it for publication. I genuinely hope that the recommendations offered will prove valuable to the authors in their efforts to publish this work.
I would suggest that the authors construct a table to clearly present the pharmacokinetic (PK) parameters, such as the maximum serum concentration (Cmax), time to reach maximum concentration (Tmax), and area under the curve (AUC), for both sulfamethoxazole and trimethoprim. This will provide a visual representation of the PK data and enhance the readability of the manuscript.
Furthermore, it would be valuable for the authors to calculate additional PK parameters, such as the elimination rate constant (Kel), elimination half-life (T1/2), and the bioavailability (F). These parameters will provide a more comprehensive understanding of the drug's pharmacokinetics and its potential efficacy in broiler chickens infected with the avian pathogenic strain of Escherichia coli.
It is evident from Figure 1 that the pharmacokinetic curve depicting the plasma concentration of trimethoprim and sulfamethoxazole shows a broad standard deviation. This wide variability raises concerns about the reliability and consistency of the results obtained. The authors should address the issue of the broad standard deviation observed in Figure 1, as it indicates a significant variability in the plasma concentration of both trimethoprim and sulfamethoxazole. This variability may have implications for the accuracy and reproducibility of the experimental findings and should be thoroughly discussed and addressed in the manuscript.
Emphasizing the use of 114 field isolates is crucial in assessing the effectiveness of trimethoprim with sulfamethoxazole as an antimicrobial. By including a diverse range of isolates from different times and spaces, we ensure that the susceptibility to antimicrobials is accurately representative of a larger population. This approach helps to avoid bias towards specific strains or regions and provides a more comprehensive understanding of the antimicrobial efficacy. The authors should highlight the importance of these 114 field isolates in their study, as it strengthens the generalizability and applicability of their findings.
It is important to note that the author did not provide a descriptive statistical analysis or perform statistical comparisons among the different experimental groups in Figures 2, 3, 4, and 5, which limits the interpretation of the results. I would suggest that the author incorporate a descriptive statistical analysis and conduct appropriate statistical comparisons to enhance the robustness and reliability of the findings.
Additionally, instead of solely relying on body weight, it would be advantageous for the author to consider using parameters such as average daily gain (ADG) and feed conversion ratio (FCR) to assess the growth performance of individual chicks. ADG and FCR provide more comprehensive insights into the growth efficiency and productivity of the chickens, allowing for a more accurate evaluation of the impact of the novel pharmaceutical compound on their overall performance
I would suggest that the author rewrites the discussion (line 204-232) in a more focused manner, specifically addressing the results obtained in this study. It is important to provide a concise and relevant discussion that directly relates to the findings and implications of the research.
In regards to the discussion section (lines 245-265), it is suggested to avoid excessive explanation of previous research details. The citation will help provide a foundation for the study's findings without overwhelming the discussion with unnecessary details.
The similarity between the results of the analytical validation and pharmacokinetic parameters of sulfamethoxazole and trimethoprim in this manuscript and the author's previous publication raises concerns about the declaration of repeated results across the two research papers. Further clarification from the author regarding the rationale and justification for presenting similar findings in both studies would be beneficial to ensure transparency and avoid redundancy in scientific literature.
It is noted that the discussion section (lines 275-289) lacks proper citations, which compromises the credibility and academic rigor of the study. To address this concern, it is recommended to incorporate appropriate citations to support the repeated restatements of the results, ensuring a stronger foundation for the study's findings.
The author's conclusion regarding the effective experimental dose of a fixed 4:1 combination for repeated dosing (20 mg in combination with 16 mg sulfamethoxazole + 4 mg trimethoprim per 1 kg b.w. of broilers) should be further clarified. It is unclear whether the author considered the principles of Target Attainment of Pharmacokinetic and Pharmacodynamics, plasma concentration above the MIC throughout the dosing interval, or classified Co-Trimoxazole as time or concentration-dependent antimicrobial. Additional information is needed to understand the basis for determining the proposed effective dose.
the authors also recommend administering the drug in four or five doses for all animals in the treated groups to ensure their survival from the infection. However, it is unclear from the provided information which specific results or data were used to calculate this frequency recommendation. It would be helpful for the authors to clarify the basis for this recommendation in order to strengthen the validity and applicability of their findings.
The authors assert that this research minimizes the selection pressure associated with antibiotic treatments, aligning with broader efforts to preserve antimicrobial efficacy in the face of resistance challenges. However, the discussion highlights (line 225-230) that the combination of trimethoprim and sulfamethoxazole is a potent "broad spectrum" antimicrobial agent, which can contribute to the selection pressure for antibiotic resistance. This inconsistency raises concerns regarding the accuracy and coherence of the author's claims, warranting further investigation into the potential impact of their research.
The conclusion section of this manuscript could benefit from more clarity. It is essential to clearly state the primary findings resulting from the significant content acquired, along with practical suggestions for its application. Additionally, providing a recommendation for further research at the end of the conclusion section would effectively encourage future investigations in this area.
The author's use of a t-test to compare the mean body weights of the broilers at the beginning and end of the experiment is questionable. A paired t-test would have been more appropriate in this case. Additionally, the absence of information regarding a normality test raises uncertainty about whether parametric or non-parametric statistics should have been used in this study. These methodological concerns warrant further clarification and potential reevaluation of the statistical analysis.
The references cited in this manuscript provide support for the statements made. However, it is recommended to include additional recent references to ensure the information remains up to date. Since, the manuscript has been cited in 36.84% (14/38) of the most recent publications within the past five years.
Author Response
Response to Reviewer 4 Comments:
Thank you very much for taking the time to review this manuscript and your prompt response!
I appreciate the opportunity to review the manuscript titled "Managing experimental infection in broiler chickens caused by an avian pathogenic strain of Escherichia coli using a novel pharmaceutical compound containing trimethoprim and sulfamethoxazole" for potential publication in Antibiotic. While the manuscript is interesting, I have significant concerns that need to be addressed in order to consider it for publication. I genuinely hope that the recommendations offered will prove valuable to the authors in their efforts to publish this work.
I would suggest that the authors construct a table to clearly present the pharmacokinetic (PK) parameters, such as the maximum serum concentration (Cmax), time to reach maximum concentration (Tmax), and area under the curve (AUC), for both sulfamethoxazole and trimethoprim. This will provide a visual representation of the PK data and enhance the readability of the manuscript. Furthermore, it would be valuable for the authors to calculate additional PK parameters, such as the elimination rate constant (Kel), elimination half-life (T1/2), and the bioavailability (F). These parameters will provide a more comprehensive understanding of the drug's pharmacokinetics and its potential efficacy in broiler chickens infected with the avian pathogenic strain of Escherichia coli.
- In the chapter 2.1 Validation and Pharmacokinetics, we added a reference to our previous work where PK parameters including Kel and T1/2 are presented in the form of tables.
It is evident from Figure 1 that the pharmacokinetic curve depicting the plasma concentration of trimethoprim and sulfamethoxazole shows a broad standard deviation. This wide variability raises concerns about the reliability and consistency of the results obtained. The authors should address the issue of the broad standard deviation observed in Figure 1, as it indicates a significant variability in the plasma concentration of both trimethoprim and sulfamethoxazole. This variability may have implications for the accuracy and reproducibility of the experimental findings and should be thoroughly discussed and addressed in the manuscript.
- Yes, explanatory text has been added to the Discussion.
Emphasizing the use of 114 field isolates is crucial in assessing the effectiveness of trimethoprim with sulfamethoxazole as an antimicrobial. By including a diverse range of isolates from different times and spaces, we ensure that the susceptibility to antimicrobials is accurately representative of a larger population. This approach helps to avoid bias towards specific strains or regions and provides a more comprehensive understanding of the antimicrobial efficacy. The authors should highlight the importance of these 114 field isolates in their study, as it strengthens the generalizability and applicability of their findings.
- First of all, I am very sorry for the error, the susceptibility/resistance determination to the combination of trimethoprim and sulfamethoxazole was performed in 113 field isolates of E. coli. This has been corrected in Table 1 and in the "Material and Methods" section.
- More information abut the tested isolates and their source has been added in the „Results“ and „Material and Method“ section.
It is important to note that the author did not provide a descriptive statistical analysis or perform statistical comparisons among the different experimental groups in Figures 2, 3, 4, and 5, which limits the interpretation of the results. I would suggest that the author incorporate a descriptive statistical analysis and conduct appropriate statistical comparisons to enhance the robustness and reliability of the findings.
- Yes, we agree. All original Figures 2,3,4 and 5 have been reworked into one Figure 2 using descriptive statistics. We have added the data set of measured b.w. with descriptive statistics to the supplementary file.
Additionally, instead of solely relying on body weight, it would be advantageous for the author to consider using parameters such as average daily gain (ADG) and feed conversion ratio (FCR) to assess the growth performance of individual chicks. ADG and FCR provide more comprehensive insights into the growth efficiency and productivity of the chickens, allowing for a more accurate evaluation of the impact of the novel pharmaceutical compound on their overall performance.
- Yes, we agree. The ADG has been added and graphically displayed in the new Figure 3. We have added the data set of calculated ADGs with descriptive statistics to the supplementary file.
I would suggest that the author rewrites the discussion (line 204-232) in a more focused manner, specifically addressing the results obtained in this study. It is important to provide a concise and relevant discussion that directly relates to the findings and implications of the research.
- Thank you for offering the possibility to improve our manuscript. Given that this introductory part of the discussion provides relevant and important information on the subject of antibiotic resistance, its legislative regulation and further information on the spectrum of the effectiveness of the tested substances, we do not consider it appropriate to omit this information in the discussion. However, we enriched the whole discussion with more focused specific comments related to the study we conducted and its results, as suggested by the reviewer.
In regards to the discussion section (lines 245-265), it is suggested to avoid excessive explanation of previous research details. The citation will help provide a foundation for the study's findings without overwhelming the discussion with unnecessary details.
- Based on your comments, we have revised this part of the manuscript focusing only on essential information while retaining the reference that directs readers to more detailed aspects of the study.
The similarity between the results of the analytical validation and pharmacokinetic parameters of sulfamethoxazole and trimethoprim in this manuscript and the author's previous publication raises concerns about the declaration of repeated results across the two research papers. Further clarification from the author regarding the rationale and justification for presenting similar findings in both studies would be beneficial to ensure transparency and avoid redundancy in scientific literature.
- We consider the publication of selected analytical validation parameters of the method used and the main PK parameters in this manuscript to be the primary reason for publication for the reader, for a good understanding of the complex PK/PD relationship of the drug and its potential efficacy in broiler chickens infected with an avian pathogenic strain of Escherichia coli. Furthermore, to avoid duplicate publication, we have comprehensively recast the PK curve in the context of this manuscript for better understanding of the readers of the journal Antibiotics.
It is noted that the discussion section (lines 275-289) lacks proper citations, which compromises the credibility and academic rigor of the study. To address this concern, it is recommended to incorporate appropriate citations to support the repeated restatements of the results, ensuring a stronger foundation for the study's findings.
- General information on dosing times of trimethoprim/sulfamethoxazole for broilers is provided in the included reference, i.e. EMA (2008). More specifically, this information can be also found in the package leaflet for specific preparations, e.g. here: https://www.noahcompendium.co.uk/?id=-463619 or here: https://www.vmd.defra.gov.uk/productinformationdatabase/files/SPC_Documents/SPC_1237476.PDF
The author's conclusion regarding the effective experimental dose of a fixed 4:1 combination for repeated dosing (20 mg in combination with 16 mg sulfamethoxazole + 4 mg trimethoprim per 1 kg b.w. of broilers) should be further clarified. It is unclear whether the author considered the principles of Target Attainment of Pharmacokinetic and Pharmacodynamics, plasma concentration above the MIC throughout the dosing interval, or classified Co-Trimoxazole as time or concentration-dependent antimicrobial. Additional information is needed to understand the basis for determining the proposed effective dose.
- Yes, we agree. We have added and clarified the text in chapter Discussion section.
The authors also recommend administering the drug in four or five doses for all animals in the treated groups to ensure their survival from the infection. However, it is unclear from the provided information which specific results or data were used to calculate this frequency recommendation. It would be helpful for the authors to clarify the basis for this recommendation in order to strengthen the validity and applicability of their findings.
- Thank you for this comment. The recommendation to administer 4 or 5 doses of the drug to increase the success rate of treatment is only our hypothesis, which must be subsequently verified or refuted by further experiments. This is explained in the "Discussion" section and supported by added citations.
The authors assert that this research minimizes the selection pressure associated with antibiotic treatments, aligning with broader efforts to preserve antimicrobial efficacy in the face of resistance challenges. However, the discussion highlights (line 225-230) that the combination of trimethoprim and sulfamethoxazole is a potent "broad spectrum" antimicrobial agent, which can contribute to the selection pressure for antibiotic resistance. This inconsistency raises concerns regarding the accuracy and coherence of the author's claims, warranting further investigation into the potential impact of their research.
- Precisely because the combination of trimethoprim/sulfamethoxazole is a broad-spectrum antibiotic that leads to the deepening of antibiotic resistance, our study sought to minimize the selection pressure by experimenting with a lower dose of these agents and found that even at a lower dose similar efficacy is maintained. Thus, this study experimentally supported the necessity of reducing the consumption of antibiotics, which is needed to prevent further development of antibiotic resistance. Given this reviewer’s comment, the discussion has been edited to avoid further misunderstanding and to keep the idea of our study clear and easy to follow.
The conclusion section of this manuscript could benefit from more clarity. It is essential to clearly state the primary findings resulting from the significant content acquired, along with practical suggestions for its application. Additionally, providing a recommendation for further research at the end of the conclusion section would effectively encourage future investigations in this area.
- Based on your comments and suggestions for improvement, we modified the conclusion of the submitted manuscript.
The author's use of a t-test to compare the mean body weights of the broilers at the beginning and end of the experiment is questionable. A paired t-test would have been more appropriate in this case. Additionally, the absence of information regarding a normality test raises uncertainty about whether parametric or non-parametric statistics should have been used in this study. These methodological concerns warrant further clarification and potential reevaluation of the statistical analysis.
- This reviewer's objection comes with a misunderstanding of the text. We correctly used the t-test when testing b.w. between independent groups (1 vs 2, 1 vs 3, and 2 vs 3) over time at the beginning and end of the experiment. The corresponding text on the use of the t-test and normality test has been clarified and added in chapter 4.6 Statistical analysis.
The references cited in this manuscript provide support for the statements made. However, it is recommended to include additional recent references to ensure the information remains up to date. Since, the manuscript has been cited in 36.84% (14/38) of the most recent publications within the past five years.
- Since not many studies are currently focused on this problem, and even more so when it comes to poultry, we focused our study primarily on this topic. However, this is associated with a lack of adequate up-to-date references. Even so, we have added several references that were published in the last five years at the latest and are at least partially related to this issue.
Thank you for all yours comment.
Please note that the manuscript was also edited based on comments from three other reviewers, so the revised version has more changes than you requested.
Round 2
Reviewer 1 Report
Comments and Suggestions for Authors
1. Line 211. In vitro should be written in italics.
2. Line 290. Staphylococcus aureus, when used repeatedly, can be abbreviated "S. aureus".
3. Line 309. The abbreviation «MICs» is not used for the first time, the decipherment of the abbreviation should be given in the first mention.
4. Line 451. "Traditional drug" or "dosage"?
Author Response
Response to Reviewer 1 Comments- Round 2:
Thank you very much for taking the time to review this manuscript and your prompt response!
- Line 211. In vitroshould be written in italics.
- In vitro has been corrected and written in italics.
- Line 290. Staphylococcus aureus, when used repeatedly, can be abbreviated "S. aureus".
- Staphylococcus aureus has nbeen abbreviated „ aureus“.
- Line 309. The abbreviation «MICs»is not used for the first time, the decipherment of the abbreviation should be given in the first mention.
- It has been corrected.
- Line 451. "Traditional drug" or "dosage"?
- „Traditional drug“ has been corrected to „traditional dossage“.
Thank you for all yours comment.
Please note that the manuscript was also edited based on comments from three other reviewers, so the revised version has more changes than you requested. Changes in the manuscript in the second round of revisions are marked in green.
Reviewer 2 Report
Comments and Suggestions for Authors
Thank you the authors for addressing all my comments.
Author Response
Response to Reviewer 2 Comments- Round 2:
Thank you very much for taking the time to review this manuscript and your prompt response and yours comments and recommendations during review process.
Please note that the manuscript still had to be revised based on the requests of other reviewers. Changes in the manuscript in the second round of revisions are marked in green.
Reviewer 3 Report
Comments and Suggestions for Authors
Thank you for your response. I agree with most of the answers.
Author Response
Response to Reviewer 3 Comments- Round 2:
Thank you very much for taking the time to review this manuscript and your prompt response and yours comments and recommendations during review process.
Please note that the manuscript still had to be revised based on the requests of other reviewers. Changes in the manuscript in the second round of revisions are marked in green.
Reviewer 4 Report
Comments and Suggestions for Authors#2 and #3 Advise the author to clearly outline the unique contributions and novel aspects of the current manuscript's PK parameter presentation, highlighting its differences from previous work.
#5 Considering the potential bias towards specific strains due to multiple isolates from the same farm and same time, I suggest that the author address this concern by providing a clear rationale for including these isolates in the study. Additionally, it would be beneficial for the authors to discuss how they mitigated the potential bias and its impact on the generalizability of their findings. Furthermore, providing a detailed analysis of the similarities and differences among isolates from the same farm and same period could enhance the transparency and credibility of the study.
#10 It's important to address the issue of potential redundancy and ensure that the publication provides unique and valuable contributions to the scientific literature. I would recommend suggesting to the author that they clearly highlight the novel aspects and contributions of the current manuscript compared to their previous publication, emphasizing the unique insights and findings that differentiate the two works. Encouraging the author to provide a thorough discussion of the distinct contributions of the current manuscript will help to address concerns about potential redundancy and plagiarism.
#12 It's important to clarify the relationship between the maximum concentration (Cmax) and the minimum inhibitory concentration (MIC) for sulfamethoxazole and trimethoprim to assess the achievement of pharmacokinetic/pharmacodynamic (PK/PD) targets. This will help ensure a thorough understanding of the rationale behind the proposed effective dose and its alignment with PK/PD targets.
#13 It would be beneficial to explicitly outline the rationale or preliminary evidence that led to the formulation of your hypothesis, enhancing the scientific basis for your recommendation. Additionally, acknowledging regulatory and safety considerations related to the use of medications in food-producing animals, and discussing relevant regulations or guidelines, will ensure the ethical and responsible use of medications in the context of the study. These additions will strengthen the validity and applicability of your research findings while addressing important regulatory and safety concerns.
Author Response
Response to Reviewer 4 Comments- Round 2:
Thank you very much for taking the time to review this manuscript and your prompt response and yours comments and recommendations during review process.
#2 and #3 Advise the author to clearly outline the unique contributions and novel aspects of the current manuscript's PK parameter presentation, highlighting its differences from previous work.
- In the Introduction section, the aims of study have been reworded to better capture the intent and difference from the previous work.
#5 Considering the potential bias towards specific strains due to multiple isolates from the same farm and same time, I suggest that the author address this concern by providing a clear rationale for including these isolates in the study. Additionally, it would be beneficial for the authors to discuss how they mitigated the potential bias and its impact on the generalizability of their findings. Furthermore, providing a detailed analysis of the similarities and differences among isolates from the same farm and same period could enhance the transparency and credibility of the study.
- In conjunction with this comment, information has been added to the Results section about MIC50 and MIC90 values, and to Material and Methods section have been added references to the methodology for collecting coli isolates for MIC determination and for determining E. coli pathotypes. In the Discussion section, there is mentioned the fact that the results of AST of isolates from the Czech Republic cannot be generalized and that for successful antibiotic treatment, prior susceptibility testing to the antibiotics considered for treatment of the infection is needed. This is generally true.
#10 It's important to address the issue of potential redundancy and ensure that the publication provides unique and valuable contributions to the scientific literature. I would recommend suggesting to the author that they clearly highlight the novel aspects and contributions of the current manuscript compared to their previous publication, emphasizing the unique insights and findings that differentiate the two works. Encouraging the author to provide a thorough discussion of the distinct contributions of the current manuscript will help to address concerns about potential redundancy and plagiarism.
- Based on the proposed modifications of this manuscript, these comments (#2, #3, #10) were accepted and incorporated into the text of the manuscript. In particular, we modified and supplemented the Introduction chapter, where we defined ourselves precisely concerning our previous work and clearly highlighted this newly presented work.
- To further clarify the potential redundancy, we can add an example of a potential reviewer contradiction: the potential reader of Antibiotics does not need to know all of the details of the analytical method applied, which are central to our previous study, however, needs to know at least the basics of the validation parameters obtained. These are important for the reader to understand the context of the newly presented work and therefore, after careful consideration, we have included them in the submitted study as well.
- We also need to define ourselves and provide an explanation for the potential plagiarism: the author collective for both papers is identical, with the only difference being that the author Kristina Putecova got married and adopted the name Tosnerova. For that reason, we have corrected her name in the author collective to Kristina Putecova-Tosnerova. Furthermore, we must note that we cite all sources in the submitted study accurately and according to established conventions. Therefore, concerning plagiarism, we carefully reviewed, revised and corrected all texts that could be subject to possible problems with plagiarism in the submitted manuscript. For the reasons described above, we must strongly reject "plagiarism" in our work, which we also unequivocally oppose.
#12 It's important to clarify the relationship between the maximum concentration (Cmax) and the minimum inhibitory concentration (MIC) for sulfamethoxazole and trimethoprim to assess the achievement of pharmacokinetic/pharmacodynamic (PK/PD) targets. This will help ensure a thorough understanding of the rationale behind the proposed effective dose and its alignment with PK/PD targets.
- Based on the reviewer's recommendation, we have included clarification of this in Chapter 3 - Discussion and Chapter 4.5 - Animal experiment.
#13 It would be beneficial to explicitly outline the rationale or preliminary evidence that led to the formulation of your hypothesis, enhancing the scientific basis for your recommendation. Additionally, acknowledging regulatory and safety considerations related to the use of medications in food-producing animals, and discussing relevant regulations or guidelines, will ensure the ethical and responsible use of medications in the context of the study. These additions will strengthen the validity and applicability of your research findings while addressing important regulatory and safety concerns.
- Based on the reviewer‘s comments, a clearly intended hypothesis of the presented study was included in the introduction, and in connection with this, the wording of the study's objectives was additionally modified to help not only the reviewer, but above all the final reader to define the subject of the study more clearly.
- Furthermore, the introduction of the manuscript was supplemented with regulatory and safety considerations based on legislation related to this issue. We had previously included more on the legislative documents related to the presented study at the beginning of the discussion, therefore we consider this addition to be sufficient.
Thank you for your all comments. Thank you for all your comments. We acknowledge all of them and hope that we have provided sufficient comments and explanations on all of them, either directly in the manuscript or in this letter. Changes in the manuscript in the second round of revisions are marked in green.
Round 3
Reviewer 4 Report
Comments and Suggestions for Authors#2, #3 and #10, Thank you for addressing the concerns regarding potential plagiarism in the manuscript. Your clarification on this matter may help to ensure the integrity of the research.
#12 and #13, In light of the recommendation for administering the drug in four or five doses for all animals in the treated groups (in the conclusion), it would be beneficial to have a detailed explanation of how the calculation to assess the achievement of PK/PD targets was derived. Asking for a reference that supports the predictive value of Cmax being 1.5 times higher than MIC would help strengthen the rationale behind the proposed effective dose. Furthermore, it would be important to assess whether the recommended dosage regimen aligns with the safety regulations for chicken and consumer health, and whether it complies with the drug label and national laws.
Author Response
Response to Reviewer 4 Comments- Round 3:
Thank you very much for taking the time to review this manuscript and your prompt response and yours comments and recommendations during review process.
#2, #3 and #10, Thank you for addressing the concerns regarding potential plagiarism in the manuscript. Your clarification on this matter may help to ensure the integrity of the research.
- Of course we agree, it's our opinion too. We assume that we do not need to add any further clarification to the manuscript in this context.
#12 and #13, In light of the recommendation for administering the drug in four or five doses for all animals in the treated groups (in the conclusion), it would be beneficial to have a detailed explanation of how the calculation to assess the achievement of PK/PD targets was derived. Asking for a reference that supports the predictive value of Cmax being 1.5 times higher than MIC would help strengthen the rationale behind the proposed effective dose. Furthermore, it would be important to assess whether the recommended dosage regimen aligns with the safety regulations for chicken and consumer health, and whether it complies with the drug label and national laws.
- We agree, but unfortunately, the recommendation of the 4th and 5th dose of the drug is only our hypothesis, which will be further tested in subsequent studies, which has been corrected in the manuscript in Conclusion and marked in gray. The predictive value of Cmax being 1.5 times higher than MIC was determined based on our experience and was based on established breakpoints of sensitivity, as already explained. The appropriateness of our proposed the predictive value of Cmax has just been verified in the preclinical study conducted. The co-authors of the study from the commercial firm Tekro plan to conduct a follow-up larger scale experiment in a good laboratory practice regime, which will be designed to answer questions on safety for animals and meat consumers and other obligations based on national veterinary drug registration regulations. In addition, the legislative requirements for consumer safety certification are based on the determination of MRLs in edible tissues of food animals and not in blood. A completely different study must deal with that.
Thank you for all your comments.